# Exploring Physical Activity Levels in Patients with Cardiovascular Disease—A Preliminary Study

**DOI:** 10.3390/healthcare12070784

**Published:** 2024-04-03

**Authors:** Saori Kakita, Takatomo Watanabe, Junya Yamagishi, Chiaki Tanaka, Daichi Watanabe, Hiroyuki Okura

**Affiliations:** 1Nursing Course, Gifu University School of Medicine, Gifu 501-1194, Japan; 2Department of Cardiology, Gifu University Graduate School of Medicine, Gifu 501-1194, Japan; watanabe.takatomo.n6@f.gifu-u.ac.jp (T.W.); okura.hiroyuki.w1@f.gifu-u.ac.jp (H.O.); 3Division of Clinical Laboratory, Gifu University Hospital, Gifu 501-1194, Japan; 4Department of Rehabilitation, Gifu University Hospital, Gifu 501-1194, Japan; yamagishi.junya.x0@f.gifu-u.ac.jp; 5Division of Nursing, Gifu University Hospital, Gifu 501-1194, Japan; tanaka.chiaki.h6@f.gifu-u.ac.jp; 6Innovative and Clinical Research Promotion Center, Gifu University Hospital, Gifu 501-1194, Japan; watanabe.daichi.k6@f.gifu-u.ac.jp; 7Department of Pharmacy, Gifu University Hospital, Gifu 501-1194, Japan

**Keywords:** circulation disorders, physical activity levels, activity meters, cardiovascular disease, household activity, questionnaire, BMI

## Abstract

Increased physical activity may prevent disease onset and severity in individuals with cardiovascular disease. However, studies evaluating physical activity in people with cardiovascular disease are limited. This prospective observational study aimed to objectively assess the level of physical activity in patients with cardiovascular disease and determine the actual extent of physical activity in their daily lives. Participants aged 20 years or older with cardiovascular disease at a cardiology clinic were included. Physical activity was measured using an activity meter with a three-axis acceleration sensor. Overall, 58 patients were included in the study. Household activities were found to be more frequent sources of physical activity. The step count was related to age and housework, while total physical activity and household activity were related to age and work. Locomotive activity was related to sex and housework. Total physical and household activities tended to decrease with age. These findings indicate the influence of work and household chores on physical activity and suggest that physical activity may be underestimated if household activity is not also assessed. These fundamental findings may provide clinical evidence to underpin physical activity for patients with cardiovascular disease.

## 1. Introduction

The total number of patients with heart disease in Japan is 1,732,000, and the number of deaths due to heart disease is 200,000 per year [1]. Furthermore, looking at medical treatment costs by injury and disease, medical expenses for cardiovascular diseases are the highest, putting pressure on national finances [2]. Efforts to prevent serious illness and reduce mortality in patients with cardiac disease should be an important public health initiative. The acceleration of atherosclerosis is a factor in the severity and recurrence of heart disease, and may occur due to physical inactivity. Habitual physical activity and aerobic exercise are said to improve vascular endothelial function, promote systemic blood circulation, improve mitochondrial function and energy metabolism in skeletal muscle cells, increase antioxidant capacity, and maintain an anti-atherosclerotic quality composition [3]. On the other hand, an increased heart rate is associated with increased hospitalization for worsening heart failure or cardiovascular-related deaths [4]. Excessive physical activity leads to an increased heart rate. In other words, moderate physical activity is necessary to prevent the severity and recurrence of heart disease.

Physical activity refers to any bodily movement that consumes more energy than when at rest [5]. Physical activity can be categorized into two types: activities of daily living, such as work or school, housework, and daily commuting; and exercise, which is planned, continuous, and intentional during leisure time to improve physical fitness. Activities of daily living, excluding exercise, such as housework, are classified as “non-exercise physical activity”. In recent years, there has been an increased interest in the relationship between non-exercise physical activity (NEAT: non-exercise activity thermogenesis) and obesity [6]. Therefore, “non-exercise physical activity” should also be considered when assessing physical activity.

There are two ways of assessing physical activity: using questionnaires, such as the International Physical Activity Questionnaire (IPAQ), or directly measuring physical activity using equipment such as activity meters [7,8,9]. Questionnaire-based assessments are easy and cheap because they do not require any instruments. However, they rely on the participants’ memory, which can lead to recall bias and an overestimation of their physical activity levels [10,11,12]. Nevertheless, most methods for assessing physical activity in people with cardiovascular disease in Japan use questionnaires, and there are very few studies on the actual measurement of physical activity. The heart rate method and calorimetry are used to measure physical activity, but they have issues such as the need for calibration before measurement and low responsiveness when the intensity of activity changes [13].

In Japan, many studies have objectively evaluated physical activity using accelerometers. However, most of these studies have focused solely on “running and walking” and have used pedometers or uniaxial accelerometers to measure these activities [14,15,16]. This approach ignores the fact that people engage in many “non-exercise physical activities” in their daily lives, such as sitting, standing, cleaning, and doing laundry. Unfortunately, pedometers and uniaxial accelerometers can only measure horizontal displacement activity, which is running and walking, and cannot measure other movements. To accurately evaluate all activities, including those beyond running and walking, researchers should use triaxial accelerometers. In Japan, the most commonly used triaxial accelerometers are the Active Tracer AC-210 (GMS, Tokyo, Japan), Actimarker EW4800 (Panasonic Electric Works, Tokyo, Japan), and Active style proHJA-750C (Omron Healthcare, Kyoto, Japan). The Active style proHJA-750C (Omron Healthcare) has proven reliable and accurate in measuring physical activity beyond running and walking [17,18]. Unfortunately, there are very few studies that have used triaxial accelerometers to evaluate physical activity levels, including non-exercise activities, in people with cardiovascular disease [19,20].

This study aims to objectively evaluate the level of physical activity among patients with cardiovascular disease using a triaxial accelerometer and to clarify the actual amount of physical activity, including movements without horizontal displacement, among patients living in the community. This study also aims to clarify the physical activity of movements without horizontal displacement, which has not been adequately assessed before and will lead to a more precise estimate of the physical activity levels of people living in the community. The findings of this study will provide fundamental data for developing strategies to regulate physical activity levels for individuals with cardiovascular disease.

## 2. Materials and Methods

### 2.1. Study Design, Setting, and Participants

This cohort study, conducted from April to October 2022, included participants aged 20 years and older who visited the cardiology department of a general hospital in Japan. Patients were included if they were undergoing any treatment to prevent disease recurrence or progression. Exclusion criteria encompassed patients with class IV cardiac function classification (according to the New York Heart Association), with dementia, with psychiatric or orthopedic disease, or unable to manage physical activity meters [21]. NYHA class IV denotes a physical activity capacity index of 2 metabolic equivalent tasks (METs) or less, which is lower than that of ordinary walking (corresponding to 3 METs). Patients were also excluded from the study due to continually declining physical activity levels.

Cooperation and consent for the study were obtained from the relevant departmental heads at the hospital. A co-researcher selected participants from among the outpatients. The principal investigator or co-investigator provided written and oral explanations of the study’s purpose, significance, and methods to the selected patients. Written patient consent was obtained. This study was approved by the Ethics Review Committee for Medical Research of the Graduate School of Medicine at Gifu University (Approval No. 2021-A223).

Data were collected from the medical records, questionnaires, and activity meters.

### 2.2. Data Collection Procedures

#### 2.2.1. Investigation of Physical Activity

Physical activity was measured using an activity meter equipped with a 3-axis acceleration sensor (Activity Monitor Active Style Pro HJA-750C, Omron Healthcare, Kyoto, Japan). This activity meter can measure not only horizontal movements, such as walking and running, but also vertical movements, such as drying laundry, vacuuming, and lifting luggage. This enables the measurement of routine activities that cannot be performed using conventional accelerometers. The reliability of the measured values and the validity of the activity intensity have been verified in previous studies [17,18].

In this study, the amount of physical activity is expressed as “Ex: activity intensity (METS) × time (hours)”. This is because, in Japan, the Ministry of Health, Labour, and Welfare uses “activity intensity (METs) × time (hours)” as an indicator of the amount of physical activity [22]. Physical activity is defined as the sum of “locomotive activities: running/walking activities”; and “household activities: activities other than running/walking activities”. “Locomotive activity” refers to activities that involve horizontal movement among running and walking, such as slow walking, brisk walking, and jogging. In other words, any activity that includes walking is considered a “locomotive activity”, whether it is for exercise purposes or just a part of daily physical activity. Furthermore, “household activity” refers to any movement that does not involve running or walking, such as cleaning, laundry, loading and unloading luggage, and sitting. These activities are also a part of daily physical activities.

Thus, the following information was collected from the activity meter: total daily physical activity (measurement unit: EX), locomotive activity (measurement unit: EX), household activity (measurement unit: EX), and daily number of steps for the period of wearing the meter. The activity meter measured locomotive and household activity of three METs or more.

Participants were instructed to wear the activity meter on their waist for at least nine days, as often as possible, except during underwater activities (e.g., swimming pools and baths) and when sleeping. To prevent adjustment of physical activity levels due to awareness of the measurements, activity meters were set to display the time. At the end of the study period, patients returned the activity meters to a box set up in the hospital or sent them by mail to the principal investigator.

A total of 101 activity meters were used, and the data obtained from the activity meters were imported into the principal investigator’s PC using dedicated software (HDV-TDH-160070, Omron Healthcare, Kyoto, Japan). The activity data for each participant were consolidated into a single spreadsheet using Microsoft Excel for Microsoft 365 MSO (ver. 2311) (Microsoft Corporation, Tokyo, Japan).

#### 2.2.2. Questionnaire Survey

The questionnaire survey was conducted in an outpatient waiting room with adherence to strict COVID-19 infection control measures. Questions included sex, age, height, weight, current disease, cardiac rehabilitation, occupation, and household chores. Height and weight were measured using a height and weight scale in a medical treatment room. The researcher calculated BMI (Body Mass Index). The Japan Society for the Study of Obesity classifies a BMI of 22 as the appropriate weight (standard weight) and statistically the weight at which one is least likely to become ill; a BMI of 25 or higher is classified as obese, and a BMI of less than 18.5 as underweight [23].

### 2.3. Statistical Analysis

Participant characteristics were expressed as mean ± SD for continuous variables and by frequency for categorical variables. Histograms were used to confirm that the distribution of all four physical activities was negatively skewed. The physical activity for each characteristic was summarized as median and interquartile range (IQR). To compare the physical activity between characteristic categories, the Mann–Whitney U test or the Kruskal–Wallis test was used. Cohen’s d and Cliff’s delta were also presented. Cliff’s delta aids in evaluating the effect size between two groups when analyzing non-parametric data [24,25]. Physical activity was visually presented in a box plot for each characteristic category.

Restricted cubic spline curves with a knot number of 3 were used to represent the smooth nonlinear relationship between age and physical activity. Spearman’s correlation coefficient was used to summarize the relationship between age and physical activity. The statistical software EZR version 1.55 (Saitama Medical Center, Jichi Medical University, Saitama, Japan) and R version 4.2.2 (R Foundation for Statistical Computing, Vienna, Austria) were used for all analyses. The level of significance was set at *p* < 0.05.

## 3. Results

### 3.1. Participant Characteristics

Of the 105 participants, 4 were excluded from the study because of the loss or submersion of their activity meters. From the integrated data, we first extracted the data of participants who wore the device for more than 600 min per day. Next, data that included at least 2 days of data measured on weekdays and at least 1 day of data measured on holidays were extracted. Finally, the data from 58 patients were included in the analysis [26]. The participants’ backgrounds are presented in Table 1. A total of 31 (53.4%) participants in this study were female. The mean age was 70.3 ± 12.1 years, and 44 (75.9%) patients were aged over 65 years. Heart failure was the most common primary disease in 13 patients (20.6%), followed by coronary artery disease and arrhythmia in 11 patients (17.5%), and lastly, valve disease was present in 10 patients (15.9%). Importantly, with regard to the stage of heart failure, 65.5% of patients were in stage A/B. While ten patients (17.2%) were currently undergoing cardiac rehabilitation, a greater proportion of patients were not. In relation to employment, 24 people (41.4%) answered yes, half of whom were males. With respect to housework, 37 participants (63.2%) answered yes, of whom nine were male.

### 3.2. Comparison of Physical Activity and Basic Attributes

A comparison of physical activity and basic attributes is presented in Table 2 and Table 3. In this study, household activities tended to be higher than locomotive activities. The number of steps taken was significantly higher when the participant was less than 65 years old and there was no housework (less than 65 years, *p* = 0.013; no housework, *p* = 0.025). Total physical activity was significantly higher in those aged <65 years and those with work experience (age < 65 years: *p* = 0.013, with work: *p* = 0.002). Locomotive activity was significantly higher for men and those without household chores (men: *p* = 0.019, without household chores: *p* = 0.003), while household activity was considerably higher for those under the age of 65 and with work (under age 65: *p* = 0.012, with work: *p* < 0.001).

Although body mass index (BMI) did not differ significantly among each physical activity level, all physical activity items tended to be lower for those with a BMI below 18.5, compared with those above 18.5 (Figure 1). In addition, all physical activity items tended to be lower in patients with heart failure at stage C/D than in those at stage A/B (Figure 1). Specifically, household activities had lower median values compared to locomotive activities in stages C/D (Figure 1).

### 3.3. Physical Activity Level Associated with Age

Regarding physical activity and basic attributes, age was associated with three items: the number of steps taken, total physical activity, and household activities. Therefore, a nonlinear regression analysis was performed to evaluate the association between physical activity levels and age. Figure 2 shows the relationship between the physical activity levels and age. The correlation coefficients between household activity and total physical activity with respect to age were –0.433 and –0.361, respectively, with the coefficients for the main effects proving significant (*p* = 0.001 and *p* = 0.013, respectively). In contrast, the correlation coefficients between the number of steps and the locomotive activity with respect to age were –0.252 and –0.217, respectively, with the *p*-value for the coefficients of the main effect being non-significant (*p* = 0.224 and *p* = 0.114, respectively). This suggests a decline in household and total physical activities with increasing age. However, no significant correlation was found for locomotive activity, with a correlation coefficient of 0.217 (*p* = 0.114) or the number of steps, with a correlation coefficient of 0.252 (*p* = 0.244). Figure 2 also shows a nearly flat regression line for all variables, indicating no change with age.

## 4. Discussion

### 4.1. Characteristics of Physical Activity

The goal for patients with chronic diseases is 6500–8500 steps per day [27]. However, the number of steps taken by the participants in this study ranged from 2166 to 5540 steps per day, which did not reach the goal for patients with chronic disease. A study performed by Saint-Maurice et al. reported that the average daily number of steps taken by adults aged 40 years and older is approximately 9200 steps per day and that a decrease to less than 2000 steps per day increases the risk of death from cardiovascular disease by approximately 51% [28]. Therefore, physical activity interventions that enable individuals to continue their current number of steps are required to reduce the mortality risk.

The Ministry of Health, Labour and Welfare recommends that physical activity in adults be at least 23 EX/week [22]. However, the physical activity of the subjects in this study was far below this level. In terms of physical activity (EX) by sex, women had a higher total physical activity and household activity, while men had a higher locomotive activity. These results are similar to those of the Hisayama Town study [29]. In another study by Hagino et al. investigating physical activity in people with chronic heart failure using an activity meter with a three-axis acceleration sensor, and they reported that household activity was higher than locomotive activity [19]. Our study showed that housework activity tends to be higher than exercise activity in individuals with cardiovascular diseases (including heart failure), which is consistent with the results of Hagino et al. [19]. When evaluating the physical activity of people with cardiovascular disease, it is crucial to consider not only the number of steps taken and locomotive activity but also household activity. Neglecting household activity may lead to an underestimation of the actual level of physical activity. However, the degree of physical activity, reflected as the cardiac load of a person, depends on their cardiac function, and the appropriate physical activity for individuals with cardiac dysfunction is determined by the anaerobic metabolic threshold. Therefore, further research on the presence or absence of physical activity surpassing the anaerobic metabolic threshold and its associated factors is also required.

It is possible that the three-axis accelerometer used in this study does not adequately assess physical activity centered on the upper body [30]. Therefore, it is highly likely that household activities, especially those involving more upper body movement, were underestimated. The current study was not able to clarify even what types of physical activities were actually performed. However, considering the characteristics of the equipment used, it will be necessary to determine the actual physical activity content in the future.

### 4.2. Physical Activity in Relation to Basic Attributes

In this study, the total physical and household activities were associated with age and work status. In contrast, the number of steps taken and physical activity were associated with the presence or absence of household chores. We expected household activity to be associated with the presence or absence of housework, but there were no significant differences. In this study, 40% of the participants were employed, with a male predominance, especially in the no-housework group. These findings suggest that the no housework group may have included more men who had jobs; therefore, activities with vertical movements performed at work were reflected in household activities. Future research should examine the relationship between the specifics of work, housework, and physical activity. Nevertheless, to date, no study has clarified the relationship between physical activity and the presence/absence of work or housework; thus, the results of this study are novel.

As BMI is calculated by dividing the weight (kg) by the square of height, it increases with weight gain. Katou et al. investigated the association between weight gain and physical activity in patients after acute myocardial infarction using a pedometer with an accelerometer (the Lifecorder GS, Suzuken, Aichi, Japan). The number of steps taken in the weight gain group was significantly lower than that in the maintenance group [31]. In our current study, physical activity levels were similar between participants with a normal BMI and those with a high BMI. However, individuals with a low BMI had lower physical activity levels. The risk of mortality from heart disease is linked not only with a high BMI but also with a low BMI [32], indicating the need for further investigation into the relationship between physical activity and a low BMI.

Patients with heart failure in stage A/B tended to take more steps and perform more physical activity than those in stage C/D, as heart function typically declines with the progression of heart failure stages. Future research should assess the presence or absence of physical activity above the anaerobic metabolic threshold and the related factors for each heart failure stage, as well as the relationship between physical activity and a participant’s life background [33].

Finally, a decline in total physical and household activities with increasing age was noted. It has been suggested that activities involving vertical movements, such as standing, become less frequent with age. The findings of this study are important because research on the impact of age-related changes in physical activity in individuals with cardiovascular disease is scarce. In addition, previous studies have shown that both upper and lower limb muscle strength declines with age [34]. In this study, the decrease in vertical movement may contribute to the decline in lower limb muscle strength. However, further studies are needed to confirm this hypothesis.

### 4.3. Strengths and Limitations of the Study

The median number of steps, total physical activity, locomotive activity, and household activity of persons with cardiovascular disease attending outpatient clinics were 3515 steps per day and 3.32, 0.68, and 2.41 EX/day, respectively. These results suggest that excluding household activity when assessing physical activity in people with cardiovascular disease may lead to an underestimation of physical activity levels. Additionally, the findings indicate that the total physical and household activities are associated with age and work status, and the number of steps taken and physical activity are associated with the presence or absence of household chores. In contrast, the BMI and heart failure stage were not associated with physical activity. Notably, the total physical and household activities tended to decrease with age.

This study had a few limitations. The study included 58 participants, a small sample size that could contribute to the lack of association observed among physical activity, BMI, and heart failure stage. Additional studies with larger sample sizes are warranted. Additionally, anaerobic metabolic thresholds were not measured in this study. Physical activity commensurate with cardiac function is crucial for individuals with cardiovascular diseases. Therefore, additional research is needed to focus on physical activity commensurate with cardiac function. It is also believed that the environment, including factors such as the area you live in, your family structure, and the size of your home, can impact physical activity. However, although this study targeted people living in rural areas, it did not investigate elements such as family structure and house size. Given that physical activity is influenced by the environment, future research will also be necessary in relation to regional characteristics and living conditions. Finally, the activity meter used in this study may not adequately assess upper body-centered activity, which is a limitation of the device. Therefore, in the future, it would be beneficial to consider a method that can comprehensively evaluate physical activity.

## 5. Conclusions

For the first time, we quantitatively assessed the physical activity of individuals with cardiovascular disease at an outpatient clinic using a triaxial accelerometer. The strength of this study lies in its novelty, considering that few studies have quantitatively evaluated physical and household activities. In addition, no studies have clarified the relationship between physical and social activities, such as work and housework. Therefore, the findings of this study may provide clinical evidence to support physical activity intervention strategies in patients with cardiovascular disease.

## Figures and Tables

**Figure 1 healthcare-12-00784-f001:**
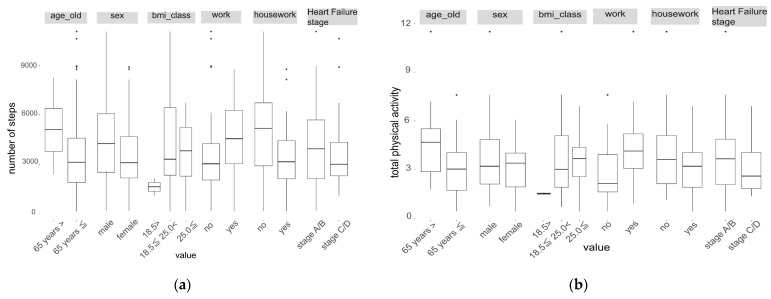
Distributions of physical activity ((**a**) number of steps, (**b**) total physical activity, (**c**) locomotive activity, and (**d**) household Activity) by participants’ characteristics. The whiskers shows from the upper and lower quartiles to the furthest points within 1.5 times the interquartile range (IQR). Data points that fall outside this range are plotted as dots.

**Figure 2 healthcare-12-00784-f002:**
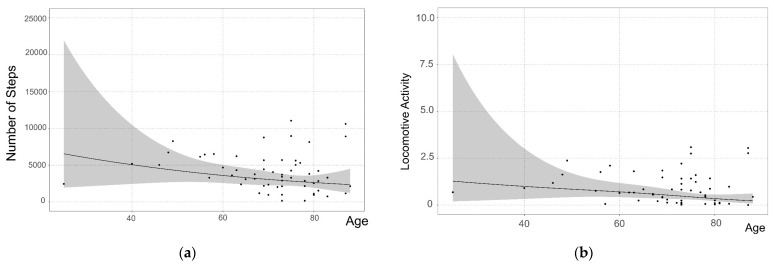
Nonlinear prediction of physical activity ((**a**) number of steps, (**b**) locomotive activity (**c**) household Activity, and (**d**) total physical activity) by age. Prediction plots were estimated using restricted cubic spline curves with a knot number of 3. The average predicted value is denoted by the solid line, and the gray shaded area shows the 95% confidence interval.

**Table 1 healthcare-12-00784-t001:** Participants’ characteristics.

Variable		Male	Female	Frequency (Percent)
Sex, N (%)		27 (46.6%)	31 (53.4%)	58 (100%)
Age (Years), N (%)	<65 years	6 (42.9%)	8 (57.1%)	14 (24.1%)
	≧65 years	21 (47.7%)	23 (52.3%)	44 (75.9%)
Height Mean (SD)	cm	165.8 (4.1)	151.7 (5.8)	158.2 (8.7)
Body Weight, Mean (SD)	kg	70.6 (13.2)	53.1 (8.7)	61.27 (14.2)
Main Disease, N (%)	Heart Failure	6 (46.2%)	7 (53.8%)	13 (20.6%)
	Coronary Artery Disease	8 (72.7%)	3 (27.3%)	11 (17.5%)
	Arrhythmia	6 (54.5%)	5 (45.5%)	11 (17.5%)
	Valve Disease	4 (40%)	6 (60%)	10 (15.9%)
	Hypertension	1 (12.5%)	7 (87.5%)	8 (12.7%)
	Cardiomyopathy	1 (20%)	4 (80%)	5 (7.9%)
	Aortic Disease	2 (66.7%)	1 (33.3%)	3 (4.8%)
	Heart Tumor	0 (0%)	1 (100%)	1 (1.6%)
	Sarcoidosis	1 (100%)	0 (0%)	1 (1.6%)
Heart Failure Stage, N (%)	A/B	19 (55.3%)	17 (44.7%)	38 (65.5%)
	C/D	8 (40%)	12 (60%)	20 (34.9%)
Cardiac Rehabilitation, N (%)	Yes	5 (50%)	5 (50%)	10 (17.2%)
	No	22 (45.8%)	26 (54.2%)	48 (82.8%)
Work, N (%)	Yes	13 (54.2%)	11 (45.8%)	24 (41.4%)
	No	14 (41.2%)	20 (58.8%)	34 (58.6%)
Housework, N (%)	Yes	9 (24.3%)	28 (75.7%)	37 (63.8%)
	No	18 (85.7%)	3 (14.3%)	21 (36.2%)

Data are presented as N (%) or mean (standard deviation).

**Table 2 healthcare-12-00784-t002:** Comparison of physical activity (number of steps, total physical activity) by participants’ characteristics.

Variable		N	Number of Steps					Total Physical Activity				
			Median	IQR	*p*-Value	Cohen’s D	Cliff’s Delta	Median	IQR	*p*-Value	Cohen’s D	Cliff’s Delta
		58	3515	(2166–5540)				3.32	(2.01–4.55)			
Age ^a^	<65 years	14	5099	(3784–6380)	0.013 *	0.55	0.44	4.69	(2.88–5.52)	0.013 *	0.90	0.44
	≧65 years	44	3124	(1900–4583)				3.04	(1.74–4.07)			
Sex ^a^	Male	27	4256	(2522–6064)	0.220	0.35	0.19	3.22	(2.13–4.86)	0.496	0.30	0.11
	Female	31	3095	(2174–4670)				3.41	(1.95–4.03)			
BMI ^b^	<18.5	2	1635	(1373–1896)	0.297	0.34	0.09	1.54	(1.51–1.57)	0.212	0.07	0.03
	18.5–25.0	33	3309	(2344–6435)				3.03	(1.92–5.11)			
	≧25.0	23	3815	(2281–5234)				3.69	(2.58–4.38)			
Work ^a^	Yes	24	4550	(3033–6268)	0.059	0.36	0.30	4.15	(3.09–5.20)	0.002 *	0.90	0.47
	No	34	3031	(2048–4247)				2.16	(1.65–3.93)			
Housework ^a^	Yes	37	3154	(2124–4434)	0.025 *	0.73	0.36	3.22	(1.92–4.07)	0.343	0.38	0.15
	No	21	5175	(2908–6723)				3.63	(2.14–5.11)			
Heart Failure Stage ^a^	A/B	38	3936	(2128–5685)	0.490	0.16	0.11	3.67	(2.10–4.88)	0.367	0.30	0.15
	C/D	20	3001	(2306–4333)				2.61	(1.84–4.07)			

* Significance: ^a^, the Mann–Whitney U test; ^b^, the Kruskal–Wallis test.

**Table 3 healthcare-12-00784-t003:** Comparison of physical activity (locomotive activity, household activity) by participants’ characteristics.

Variable		N	Locomotive Activity					Household Activity				
			Median	IQR	*p*-Value	Cohen’s D	Cliff’s Delta	Median	IQR	*p*-Value	Cohen’s D	Cliff’s Delta
		58	0.68	(0.22–1.42)				2.41	(1.41–3.37)			
Age ^a^	<65 years	14	0.84	(0.67–1.73)	0.013 *	0.33	0.28	3.07	(2.33–4.58)	0.012 *	1.00	0.44
	≧65 years	44	0.54	(0.13–1.28)				1.99	(1.26–3.17)			
Sex ^a^	Male	27	0.98	(0.44–1.69)	0.220	0.69	0.36	2.02	(1.21–3.17)	0.344	0.03	0.15
	Female	31	0.53	(0.13–0.84)				2.76	(1.49–3.34)			
BMI ^b^	<18.5	2	0.24	(0.14–0.34)	0.372	0.31	0.04	1.30	(1.23–1.37)	0.255	0.07	0.16
	18.5–25.0	33	0.67	(0.13–1.47)				2.25	(1.35–3.43)			
	≧25.0	23	0.76	(0.28–1.02)				2.78	(1.99–3.24)			
Work ^a^	Yes	24	0.88	(0.39–1.61)	0.113	0.24	0.25	3.17	(2.63–3.88)	<0.001 *	0.99	0.54
	No	34	0.54	(0.13–0.95)				1.66	(1.25–2.44)			
Housework ^a^	Yes	37	0.55	(0.13–0.85)	0.003 *	1.09	0.47	2.62	(1.44–3.48)	0.385	0.03	0.14
	No	21	1.41	(0.44–2.21)				2.19	(1.35–2.80)			
Heart Failure Stage ^a^	A/B	38	0.72	(0.25–1.46)	0.598	0.12	0.09	2.70	(1.50–3.53)	0.251	0.32	0.19
	C/D	20	0.59	(0.19–1.16)				1.66	(1.39–2.90)			

* Significance: ^a^, the Mann–Whitney U test; ^b^, the Kruskal–Wallis test.

## Data Availability

The datasets used and/or analyzed during this study are available from the author, Saori Kakita, upon reasonable request.

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
