# Peer review of "Exploring Physical Activity Levels in Patients with Cardiovascular Disease—A Preliminary Study"

_healthcare, 2024, doi:10.3390/healthcare12070784_

Round 1
Reviewer 1 Report
Comments and Suggestions for Authors
1. The article, 'Exploring Physical Activity Levels in Patients with Cardiovascular Disease' highlights an important area that is not researched on enough. This because, the role of living activities in the physical activity profile of an aging individual is nicely captured in this research. This makes it highly innovative and informative to the reader.
2. Nevertheless, the introduction (lines 38 to 39 on page 1) is confusing especially: " Furthermore, in individuals with cardiovascular disease, increased physical activity may prevent disease onset and severity".
If one has already been diagnosed with cardiovascular disease, then there is no prevention and onset. However, the authors can talk about "management of the disease and its severity". The authors should review and edit this section.
2. On page 2, lines 62-69, the authors write in the future tense when the study is already done, and we are reading a report or outcome. Once again, the authors need to edit the language.
3. Under materials and methods, the authors describe the design as prospective, observational study! Some clarification or expanded description is needed as the respondents wore an activity meter and were not directly observed doing their physical activities. Secondly, there was use of a questionnaire, so it leans towards survey. The authors should tie these approaches more coherently.
4. Check punctuation on page 4, lines 154 - 156.
5. Page 7, line 214 sounds incomplete.
Comments on the Quality of English Language
1. On page 2, lines 62-69, the authors write in the future tense when the study is already done, and we are reading a report or outcome. Once again, the authors need to edit the language.
2. Check punctuation on page 4, lines 154 - 156.
5. Page 7, line 214 sounds incomplete.
Reviewer 2 Report
Comments and Suggestions for Authors
The article entitled "Exploring Physical Activity Levels in Patients with Cardiovascular Disease" brings relevant information about public health and seems to provide important insights for future studies, but lacks care in the writing and methodology that will be addressed below.
Title
Based on the analysis of the document, I believe that this article should be considered a preliminary study, in which case I suggest that the title makes this clear. Being something like: "Exploring Physical Activity Levels in Patients with Cardiovascular Disease – A preliminary study"
Introduction
The introduction lacks a deeper understanding of the measure, better explaining the use of sensors, citing articles to base the choice on, and justifying the use for the variables analyzed later.
Line 37: Reference this statement or, if it is related to the previous sentence, merge the two sentences into a more continuous text.
Line 44: The statement "Urgent Issue" could be replaced by an important intervention for public health, adapting clearly to the context. I believe that the term would apply to medical cases of intervention in the medium term, in the case of public health eventually the interventions occur in the long term, aiming at improvements for future generations.
Line 53: It is stated that studies on the actual physical assessment are scarce, but no studies are cited, in this case reference accordingly.
Line 54: We know that measuring the actual data is more accurate than other methods, for example, measuring oxygen consumption would be more ideal through calorimetry than through estimation formulas, because the actual values reflect the current condition of the individual, who may have specific characteristics of physical activity level. I hope for this kind of complement in this excerpt, to explain why this direct measurement evaluation in this case may be better than other methods. And cite if there are studies addressing this information.
Line 55: Again, the text says "many studies", but the studies are not cited. Cite the studies used in this statement.
Materials and method
Line 111: A concern regarding the methods is the validity of the use of the waist activity sensor. Does it reflect daily physical activity? If the activity involves greater movements of the upper limbs, is he able to provide data on that activity?
If it is not possible to provide this information or if the sensor has problems with this type of measurement, I believe that greater care should be taken in writing the results and problems, since it may also be providing unreliable data, such as questionnaires or similar tools.
Participants
I felt the lack of a sample calculation, in which case it seems pertinent to provide power data for the sample evaluated, explaining the data used in the calculation, such as effect size, error, or other data provided for this. Without this data, affirming that the sample is sufficient, large or small, as it eventually appears in the discussion does not bring information based on data, but rather on opinion.
Statistical analysis
Statistical analysis is pertinent to the topic, and the use of T-test data may not be sufficient to provide information about hypotheses. In this case I suggest the use of effect sizes for comparative measurements, they can be reported as Cohen's D for example, making it clear which thresholds are adopted for each band. Without this, it becomes more difficult to make a statement about the data.
Results
Table 1: Create a table with sex data, that is, one column for female, one column for male, and one column for total data. This makes it easier to see.
Discussion
Line 257: This statement must be related to the power of the sample, as this calculation was not performed, and mainly, for this grouping we do not know the ideal N, this statement needs care. Better justify why you should think about the number of participants.
Line 262: Again, to say that the number of participants is not sufficient without a sample calculation seems wrong to me.
In this case, my concern is, if the N was not enough then this can be considered a preliminary study, which seeks to understand the problem so that further research can deepen the topic.
Author Response
Dear Editor:
We/I wish to re-submit the manuscript titled “Exploring Physical Activity Levels in Patients with Cardiovascular Disease– A preliminary study.” The manuscript ID is healthcare-2862119.
We thank you and the reviewers for your thoughtful suggestions and insights. The manuscript has benefited from these insightful suggestions. I look forward to working with you and the reviewers to move this manuscript closer to publication in the Healthcare.
The manuscript has been rechecked and the necessary changes have been made in accordance with the reviewers’ suggestions. The responses to all comments have been prepared and attached herewith/given below.
Thank you for your consideration. I look forward to hearing from you.
Sincerely,
Saori Kakita
Gifu University School of Medicine
Gifu 501-1194, Japan
Tel.: +81-58-293-3253
kakita.saori.j0@f.gifu-u.ac.jp

Reviewer 3 Report
Comments and Suggestions for Authors
Concerning the manuscript: Exploring Physical Activity Levels in Patients with Cardiovascular Disease, submitted to Healthcare. It is a good work with sound methodology and interesting results, but improvements are needed. With the utmost respect, allow me to give you a few suggestions.
The great value of the study is in measuring physical activity by acceleration sensors. In fact, one caveat for the use of questionnaire is that they cannot capture all of the nuances of day-to-day life and do not reflect with precision real-life situations. The authors must be more emphatic in describing the importance of measuring correctly the behavior of individuals in their natural habitat, avoiding disruptions of daily routines.
The soul of introduction was to establish the importance of accurate method in estimating physical activity, but the reasons for studying populations with cardiovascular disease doesn't look convincing. The authors do not explain the reason why they decided to explore this population. Excluding the justification related to cardiovascular mortality, why emphasis is placed on cardiovascular disease? It is necessary to emphasize the mechanisms underlying the physical inactivity and cardiovascular diseases (e.g. reduced energy expenditure > fat accumulation> inflammation> insulin resistance> dyslipidemia> LDL modifications and increase in endothelial permeability, and finally atherosclerosis).
Line 103: Locomotive activity refers to activities with parallel movements, such as slow to fast walking and jogging, while household activity refers to activities involving vertical movements, such as cleaning, doing laundry, and lifting and lowering luggage”. The concepts are not well referenced. In my opinion, the authors don't seem to be discriminating the engagement in:
1) volitional activities (e.g. time spent in sports or planned exercise),
2) nonvolitional activities (scope of spontaneous physical activity - SPA). It must be remembered that SPA is not only associated to movements associated with displacement (i.e. ambulation), but also includes activities without horizontal displacement such as spontaneous muscle contractions, posture maintenance, fidgeting.
Please see these studies:
https://pubmed.ncbi.nlm.nih.gov/21177942/
https://www.science.org/doi/abs/10.1126/science.283.5399.212
https://www.science.org/doi/abs/10.1126/science.1106561
https://onlinelibrary.wiley.com/doi/abs/10.1111/j.1365-201X.2005.01467.x
The limitation of the study regarding the concepts of PA is missing and has to be included. I would like to know the authors' line of reasoning about this.
Taking into consideration that the environment is the most important factor that influences physical activity, I suggest the inclusion of an additional discussion about home conditions (home space, number of individuals housed in a same house, cage urban vs rural setting). Inclusion of data, if you have, would highly acceptable. Questions such as urban danger, industrialization may be acting in inhibiting physical activity. By this reason, rural (or green space areas) are supposed to increase physical activity in comparison to urbans space. Evidence from animal studies shows that a large cage stimulates a physically active lifestyle. Please see these studies:
https://www.sciencedirect.com/science/article/pii/S0361923021002240
https://journals.sagepub.com/doi/abs/10.1177/00236772211065915

Author Response

(The authors gave the same response as above.)

Round 2
Reviewer 2 Report
Comments and Suggestions for Authors
The article ""Exploring Physical Activity Levels in Patients with Cardiovascular Disease" " shows essential informations on the subject, exploring contents that are little addressed in the literature, especially in an associated way. The topic of this study is interesting and it provides important findings. In general, the reviewers' recommendations were followed or clarified according to the requests. So, I recommend the publication of this manuscript in JFMK.
Author Response
We thank you for your kind remarks and insights. This manuscript benefited from these insightful remarks.
Our paper, "Exploring Physical Activity Levels in Patients with Cardiovascular Disease," explores a topic rarely addressed in the literature in a particularly relevant way. Thank you for recommending this paper.
Reviewer 3 Report
Comments and Suggestions for Authors
line 127: Locomotive activity” refers to activities that involve horizontal movement among running and walking. What references are you using to define this? I want to see the studies that said this.
A jump or squat (which does not involve horizontal displacement) then could they not be considered locomotive activities? The definition does not seem to correctly contemplate all situations.
Cliffs delta must be explained in statistical section.
Author Response
Thank you for your careful peer review. Please see the attached revised version with improvements, as well as our responses below. Please review the corrections you have made to improve the paper.
